# Evolutionary selection of proteins with two folds

**Joseph W. Schafer[1] & Lauren L. Porter [1,2] ✉**

Although most globular proteins fold into a single stable structure, an increasing number have been shown to remodel their secondary and tertiary structures in response to cellular stimuli. State-of-the-art algorithms predict that these fold-switching proteins adopt only one stable structure, missing their functionally critical alternative folds. Why these algorithms predict a single fold is unclear, but all of them infer protein structure from coevolved amino acid pairs. Here, we hypothesize that coevolutionary signatures are being missed. Suspecting that single-fold variants could be masking these signatures, we developed an approach, called Alternative Contact Enhancement (ACE), to search both highly diverse protein superfamilies–composed of single-fold and fold-switching variants–and protein subfamilies with more fold-switching variants. ACE successfully revealed coevolution of amino acid pairs uniquely corresponding to both conformations of 56/56 fold-switching proteins from distinct families. Then, we used ACE-derived contacts to (1) predict two experimentally consistent conformations of a candidate protein with unsolved structure and (2) develop a blind prediction pipeline for fold-switching proteins. The discovery of widespread dual-fold coevolution indicates that fold-switching sequences have been preserved by natural selection, implying that their functionalities provide evolutionary advantage and paving the way for predictions of diverse protein structures from single sequences.

Though machine learning methods have recently revolutionized protein structure prediction[1–3], some classes of proteins remain a challenge[4–7]. For example, fold-switching proteins[8], also known as metamorphic proteins[9], transition between two sets of stable secondary and tertiary structure[8,10]. These structural transitions modulate protein functions involved in suppressing human innate immunity during SARS-CoV-2 infection[11], controlling the expression of bacterial virulence genes[12], maintaining the cycle of the cyanobacterial circadian clock[13,14], and more[15,16]. Despite their biological importance, AlphaFold2 predicts only one conformation for 92% of known dual-folding proteins, and 30% of the predicted conformations were likely not the lowest energy state[17]. Other structure prediction algorithms, such as trRosetta[18] and EVCouplings[19], also systematically failed to predict experimentally validated fold switching in the universally conserved NusG family of transcription factors[15,20].

Most state-of-the-art protein structure prediction algorithms, including all just mentioned, infer folding information from evolutionary conservation patterns. Very early studies of protein structure[21] recognized that covarying amino acid pairs in homologous sequences, also known as coevolved residue pairs, tend to be in direct contact[22,23]. These coevolved contacts can greatly constrain the number of possible conformations that computational methods must sample to predict a protein's fold[24], motivating the development of increasingly sophisticated methods that infer amino acid coevolution[25–31]. Multiple sequence alignments (MSAs), collections of sequences homologous to the sequence of interest, are the inputs to most of these methods. Typically, the accuracy of inferred coevolved residue pairs increases

[1]National Library of Medicine, National Center for Biotechnology Information, National Institutes of Health, Bethesda, MD 20894, USA. [2]National Heart, Lung, and Blood Institute, Biochemistry and Biophysics Center, National Institutes of Health, Bethesda, MD 20892, USA. ✉e-mail: lauren.porter@nih.gov

with MSA depth[30], though recent deep learning-based methods can make accurate inferences from shallow MSAs[3,31].

The heavy reliance of structure prediction algorithms on coevolutionary information suggests two possible explanations for the lack of predicted fold-switching proteins: (1) fold-switching proteins are rare, transient evolutionary byproducts that bridge two distinct folds but are not selected to assume distinct conformations[32,33], or (2) both conformations of fold-switching proteins are selected, but current prediction strategies unintentionally miss the evolutionary signatures of two folds. Frequent coevolution of both conformations, if present, both supports the idea that protein fold switching confers selective advantage[8,16,34] and provides a potential strategy to identify additional fold-switching proteins.

Some previous work hints that amino acid contacts unique to each conformation of fold-switching proteins may have coevolved, a phenomenon hereafter called dual-fold coevolution. For example, we recently identified fold-switching proteins within the universally conserved NusG transcription factor family by leveraging structural information derived from MSAs from protein superfamilies (deep MSAs containing a large clade of diverse-yet-homologous sequences) and protein subfamilies (shallow MSAs with sequences similar to a target of interest)[20]. Furthermore, Dishman and colleagues found that several reconstructed ancestors of the fold-switching chemokine XCL1 switch folds, from which they concluded that XCL1 fold switching was evolutionarily selected[34]. These studies, though suggestive, focus on a couple of specific systems and infer fold switching from experimental characterization of a few variants (XCL1) or inconsistent secondary structure predictions (NusG). Weak coevolutionary couplings of a fold-switching NusG have also been predicted, though the couplings had high proportions of noise[35].

Here, we find dual-fold coevolution in 56/56 fold-switching proteins from many diverse families. To do this, we applied unsupervised learning techniques to both superfamily and subfamily-specific MSAs of all known fold switchers with two distinct experimentally determined structures and leveraged our findings to bias AlphaFold2 to predict both conformations of a candidate fold-switching NusG protein with <30% aligned identity to both of its PDB homologs. Realizing that the information from dual-fold coevolution can facilitate predictions of two protein structures from one amino acid sequence, we developed a pipeline to blindly predict fold-switching proteins from their sequences. This pipeline correctly identified 13/56 fold-switching proteins (23%) with a false-positive rate of 0/181. Together, our results indicate that (1) fold-switching proteins have largely been selected by evolution and likely confer selective advantage and (2) the information from dual-fold coevolution can be leveraged to predict fold-switching proteins from sequence.

## Results

### Methodologies to infer and analyze residue-residue coevolution

To assess the frequency of dual-fold coevolution among unrelated fold-switching proteins, we applied unsupervised learning techniques to both superfamily and subfamily-specific MSAs of 91 fold switchers with two distinct experimentally determined structures[17]. One technique identifies coevolution of amino acid pairs using Markov Random Fields (MRFs). The MRF construction offers several advantages: (i) it converges to a global minimum as MSA depth increases, (ii) it can generate reasonable predictions from fairly shallow MSAs, and (iii) the MRF formalism accounts for noncausal correlations that arise when two residues interact with a third but not with one another[36-38]. Among the numerous MRF-based methods[19,27,39], we selected GREMLIN (Generative Regularized ModeLs of proteINs) because of its superior performance[36,37]. The second technique, MSA transformer, infers coevolved amino acid pairs using a language model that focuses on both evolutionary patterns of amino acids within an MSA (column-wise attention) and properties of the individual sequences (row-wise attention), often with better accuracy than GREMLIN for single-fold proteins[31].

We gauge the success of these methods by quantifying the overlap between predicted and experimentally determined residue-residue contacts from both folds. These comparisons are easily visualized with contact maps, which display amino acid pairs either measured or predicted to be proximal (heavy atom distance ≤8 Å[37]). Though typical contact maps are symmetric about the diagonal, those used here are asymmetric to maximize information content. For example, the large light gray circles in the upper triangular portion of Fig. 1 represent contacts unique to the experimentally determined monomeric fold of

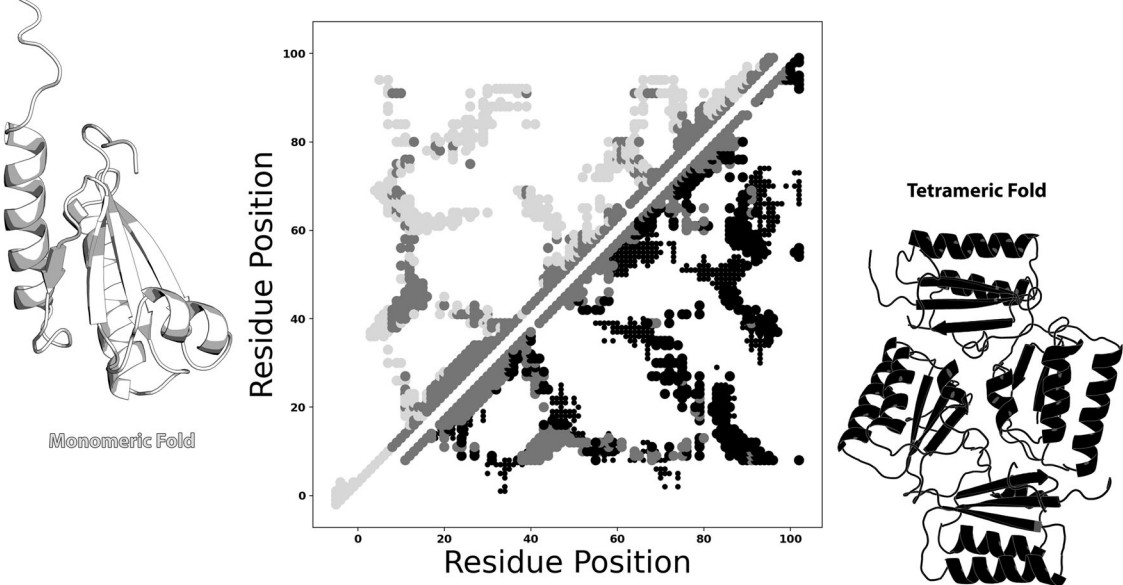

**Fig. 1 | Example of a dual fold contact map from experimentally determined structures.** KaiB monomeric/tetrameric heavy-atom contacts within 8 Å are shown in the upper/lower triangles of the contact map in light gray/black. Contacts common to both folds are shown in medium gray. Interchain contacts within 10 Å are shown as smaller circles in their respective colors. Monomeric/tetrameric contacts were calculated from PDBs 1T4Y/4KSO. Protein structures were generated with PyMOL[80]. Plots in all figures were generated with Matplotlib[81]. Source data are provided as a Source Data file.

KaiB, while the black circles in the lower triangular portion represent contacts unique to KaiB's experimentally determined tetrameric fold. Contacts common to both experimentally determined folds are shown in medium gray on both sides of the diagonal. Where appropriate, interchain contacts are represented by smaller circles using the same color scheme (in this case, black but not light or medium gray). Predicted contacts, shown in other figures below, are smaller and teal.

Correct predictions are opaque circles; incorrect predictions are translucent diamonds.

### Approach to identify dual-fold coevolution
Figure 2 depicts our workflow to search for dual-fold coevolution (Methods), called alternative contact enhancement (ACE). The query sequence, which corresponds to two distinct experimentally

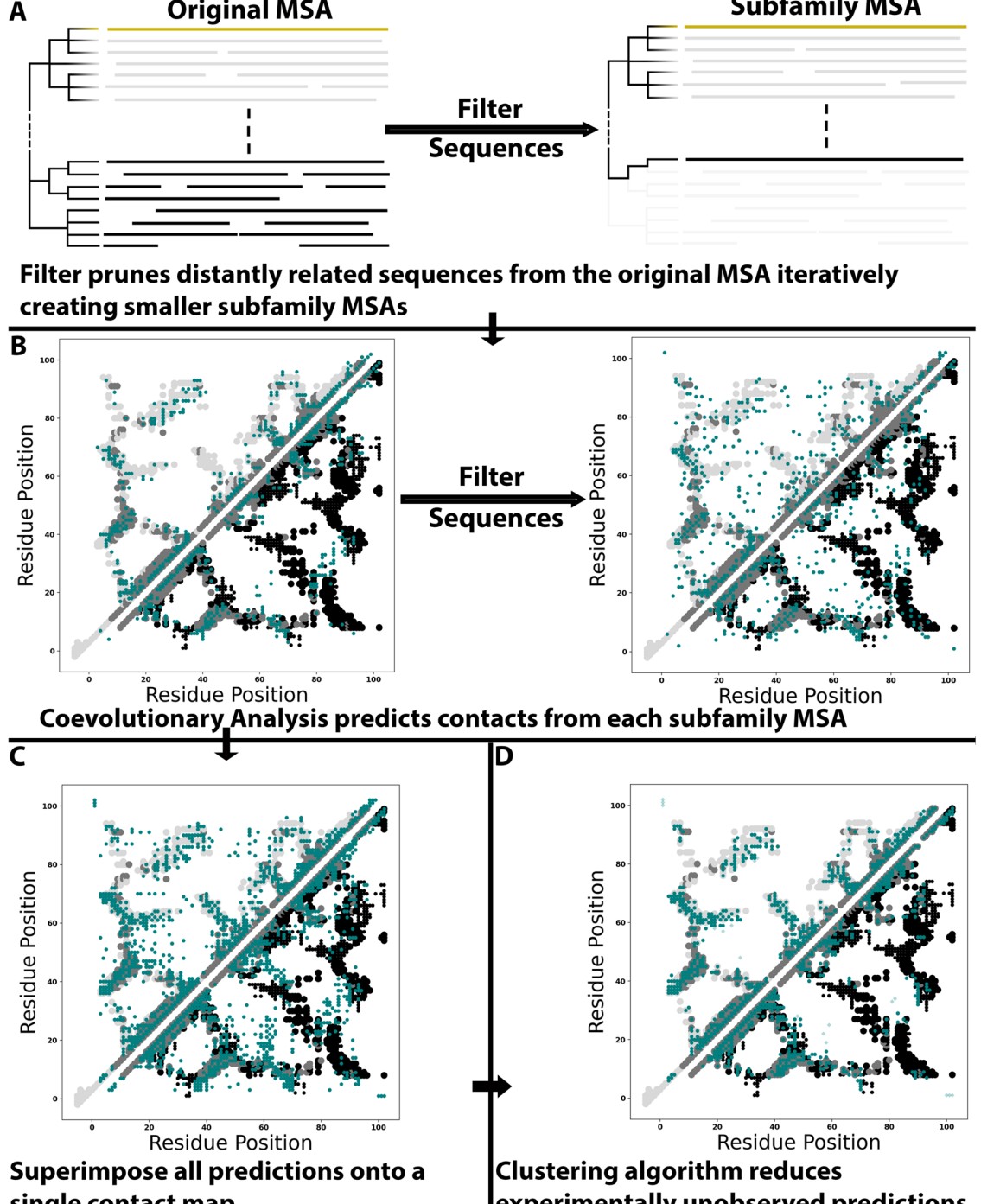

**Fig. 2 | Graphical depiction of Alternative Contact Enhancement (ACE), using KaiB as an example input. A** An MSA suitable for coevolutionary analysis is pruned by the identity of its sequences to the query sequence (yellow), removing distantly related sequences from the dataset and generating subfamily-specific MSAs. **B** Each MSA (original + all pruned) is used as input for coevolutionary analysis. **C** Predictions from all MSAs are superimposed on a single contact map. **D** A clustering algorithm filters noise, leaving dense clusters of predicted amino acid contacts. Contacts unique to the dominant/alternative folds are light gray/black; common contacts are light gray; experimentally consistent predictions are teal circles; incorrect predictions (noise) are translucent teal diamonds. Figure 1 provides an explanation of the dual-fold contact maps used here. Source data are provided as a Source Data file.

determined structures, is used to generate a deep MSA. This MSA is pruned to create successively shallower MSAs with sequences increasingly identical to the query (Fig. 2A). These increasingly subfamily-specific MSAs are intended to unmask coevolutionary couplings from alternative conformations, as they did with RfaH, a fold-switching NusG protein whose ground state α-helical conformation was identified only in subfamily-specific MSAs[20,35]. Accordingly, coevolutionary analysis is performed on each MSA using GREMLIN and MSA Transformer (Fig. 2B). Predictions from both methods run on these nested MSAs are combined and superimposed on a single contact map (Fig. 2C). Finally, these predictions are filtered by density-based scanning to remove noise (Fig. 2D). Predicted contacts are categorized as follows. Dominant fold: unique contacts corresponding to the experimentally determined structure that overlaps most with predicted contacts from the deepest MSA (light gray contacts in Fig. 2B–D); Alternative fold: unique contacts corresponding to the other experimentally determined structure (black contacts in Fig. 2B–D); Common: predicted contacts overlapping with experimentally determined contacts shared by both folds (gray contacts symmetric on both sides of the diagonal in Fig. 2B–D); Unobserved: predicted contacts that do not overlap with any experimentally determined contacts (readily visible in Fig. 2C). As shown in previous work, unobserved contacts can result from alternative conformations consistent with molecular dynamics simulations revealing folding intermediates[35] and other structural dynamics[40]. Unobserved contacts can also be erroneous (noise).

## Evolutionary selection of dual-fold proteins

We applied ACE to all known fold-switching proteins, 91 single sequences with two distinctly folded experimentally determined structures[17]. These proteins are found in all kingdoms of life and represent >80 distinct fold families (Supplementary Table 1). Although efforts were made to generate the deepest possible MSA for each fold-switching sequence (Methods), the depths of 35 MSAs were too shallow for downstream analysis (<5*length of query sequence[37]) and one displayed severe artifacting after analysis. Thus, ACE was applied only to the remaining 56 fold-switching sequences with sufficiently deep MSAs (Supplementary Table 1, Supplementary Figs. 1–10). Conformations with more contacts predicted in the superfamily MSA are denoted "dominant", and those with fewer predicted contacts, "alternative". This terminology holds no biophysical significance: 33% of "dominant" conformations do not correspond to the lowest energy states (Supplementary Table 2).

ACE predicted substantially more correct contacts than the standard approach, i.e., coevolutionary analysis run on deep superfamily MSAs alone[30]. Most notably, predicted amino acid contacts uniquely corresponding to the 56 alternative conformations were highly enhanced, with mean/median increases of 201%/187% (Fig. 3a). The number of correctly predicted contacts also increased for all 56 proteins, with mean/median increases of 111%/107% (Fig. 3b). Experimentally unobserved contacts were amplified substantially less than either alternative or correctly predicted contacts, with mean/median increases of 42%/47% (Supplementary Fig. 11). Prior to density-based filtering, mean/median unobserved contacts were amplified by 69/73%, demonstrating that, on average, 39% of the extra unobserved contacts accrued from subfamily MSAs is sparsely distributed.

Statistical analysis confirmed that the additional coevolutionary contacts identified by our approach are much more likely to be products of evolution than chance. Specifically, the likelihood of generating the additional correct contacts–with concomitant unobserved contacts–was very low for all 56 fold-switching proteins, with $p$-values ranging from 0.0091 to 0 (one-tailed hypergeometric test, Supplementary Table 1). These low $p$-values demonstrate that the dual-fold coevolutionary signatures identified by GREMLIN and MSA Transformer are significant, indicating that evolution has selected for protein

sequences that assume two distinct folds. Importantly, dual-fold coevolution was largely not observed in a test set of 181 single-fold proteins: the distribution of non-dominant contacts in this set was significantly lower than for fold switchers ($p < 1.1 * 10^{-94}$, Epps-Singleton test, Fig. 3c, Supplementary Table 1).

## Enhanced contacts originate largely from shallow subfamily-specific MSAs

We sought to identify which subfamily MSAs most enhanced predictions. For all 56 fold-switching proteins, we determined the cumulative number of alternative contacts predicted as a function of MSA depth. Correctly predicted contacts were quantified and binned by the number of sequences in the shallowest MSA normalized by the number of sequences in the original superfamily MSA. For instance, a superfamily MSA with 20,000 sequences could have smaller pruned MSAs with 19,050, 15,100, and 999 sequences, which would fall in bins 0.95, 0.70, and 0.0, respectively. The mean and standard deviation of the cumulative number of predicted contacts were calculated across all bins for each of the 56 proteins, from which the z-scores of the numbers of predicted contacts were determined. This approach allowed statistical variations in the number of predicted contacts to be compared directly between all 56 proteins despite large variations in the raw numbers of contacts predicted across families.

Many enhanced contacts originated from shallow subfamily-specific MSAs (Fig. 4, Supplementary Fig. 12). Most notably, z-scores of the numbers of alternative contacts increased sharply in subfamily-specific bins (0.00–0.15) (Fig. 4a, b). Subfamily-specific bins 0.0–0.1 had median z-scores >0, indicating more predicted contacts than expected across the 56 families, on average (Fig. 4a). Furthermore, subfamily-specific MSAs constituting <20% of their unpruned superfamily MSAs provided over half of the enhancement in predicted alternative contacts (Fig. 4b). As hypothesized, these results demonstrate that most contacts corresponding to the alternative conformations of fold-switching proteins originate from shallow MSAs with sequences most similar to the known fold-switching sequence.

A modest increase in alternative contacts was also observed upon pruning 5–10% of the least similar sequences in the deepest superfamily MSAs (bin 0.9, Fig. 4a, b), suggesting that eliminating the most dissimilar sequences may enhance overall MSA quality. Z-scores increased gradually between bins 0.85 and 0.15 (Fig. 4a). Subfamily-specific MSAs enhanced predictions of dominant and common contacts also (Supplementary Fig. 12a-d). Importantly, our noise reduction strategy preferentially eliminated experimentally unobserved contacts (Supplementary Fig. 12e, f): z-scores of the number of experimentally consistent contacts in all three categories remained essentially constant, while the z-score of the number of experimentally unobserved contacts decreased to ~0, on average (Supplementary Fig. 12g).

## Masking dominant contacts allows AlphaFold2 to predict both structures of a distant NusG homolog

Widespread dual-fold coevolution opens the possibility of predicting both conformations of a fold-switching protein from its sequence. We tested this possibility on a NusG Variant with low sequence identity (≤29%) to homologs with experimentally determined three-dimensional structures. NusG proteins are the only transcription factors known to be conserved in all kingdoms of life[41]. Unlike most NusGs with atomic level structures, whose C-terminal domains (CTDs) assume a β-roll fold, this Variant's CTD switches from an α-helical ground state to a β-roll[20], much like its homolog, RfaH[42]. Nevertheless, AlphaFold2 consistently predicts that the CTD of this Variant assumes a β-roll fold only (Fig. 5a, Supplementary Fig. 13). This prediction corroborates the observations discussed previously: all NusG CTDs are expected to assume β-roll folds (dominant conformation), though a subpopulation can also assume α-helical folds (alternative conformation). To test whether the coevolutionary

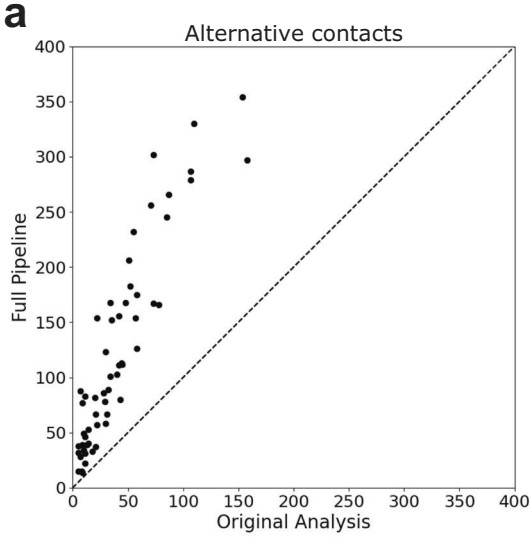

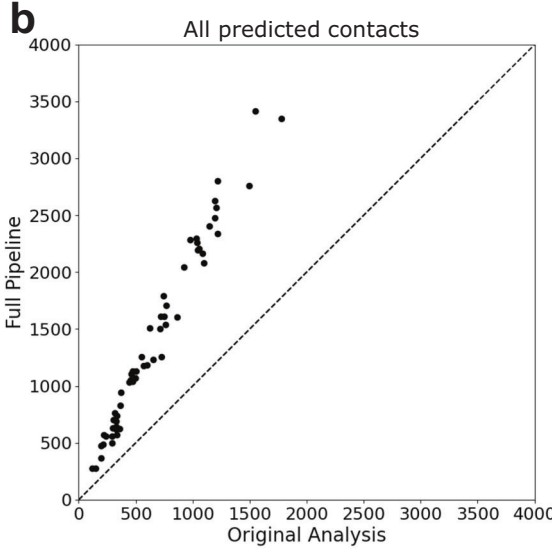

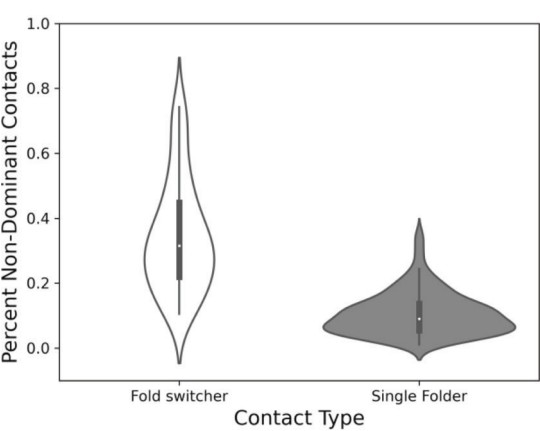

**Fig. 3 | ACE amplifies correctly predicted contacts for fold-switching proteins.** Amplification is observed for 56/56 predicted contacts uniquely corresponding to the alternative fold (**a**) and for all predicted contacts (**b**). Identity lines in both plots are dashed lines. **c** Amplification of alternative contacts occurs much more frequently in fold switchers than among single folders. Violin plots show the distributions of %non-dominant contacts for fold-switching and single-fold proteins. The left and right distributions were generated from $n = 56$ and $n = 181$ datapoints, respectively. Inner bold black boxes span the interquartile ranges (IQRs) of each distribution (first quartile, Q1 through third quartile, Q3); medians of each distribution are white dots, lower line (whisker) is the lowest datum above Q1-1.5*IQR; upper line (whisker) is the highest datum below Q3 + 1.5*IQR. Source data are provided as a Source Data file.

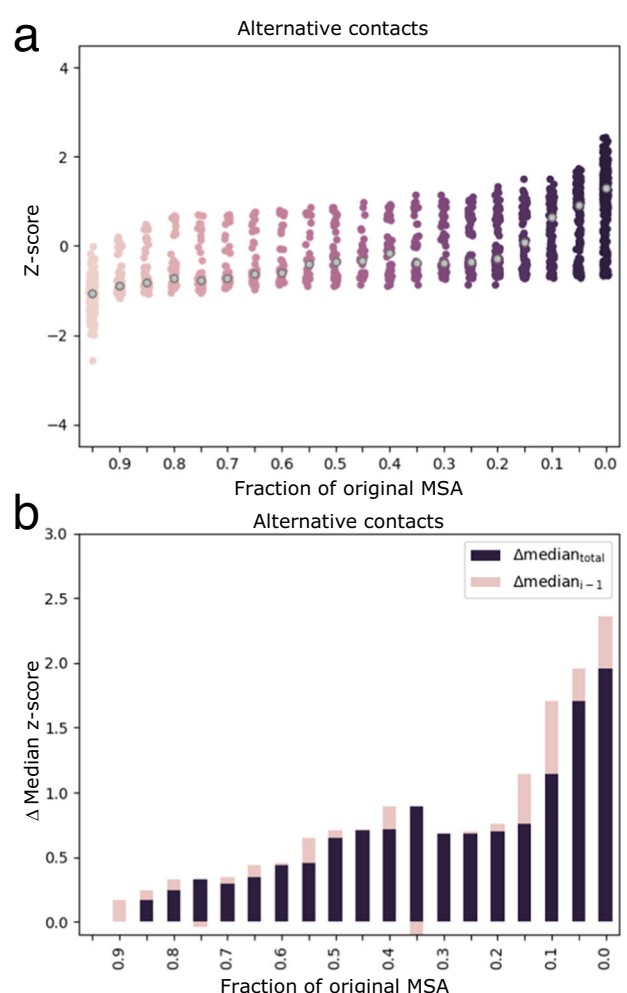

**Fig. 4 | Alternative contacts are enhanced largely by subfamily-specific MSAs.** **a** Z-scores of predicted alternative contacts increase as MSAs become shallower and more similar to the fold-switching sequence of interest. Median z-scores of each bin are gray. **b** Z-scores of predicted contacts change most in deepest and shallowest MSAs. Purple bars are differences between median z-score of bin (gray dots in (**a**)) and median z-score of the deepest MSA. Pink bars are differences between median z-score of bin and median z-score of next deepest bin. Source data are provided as a Source Data file.

signal of the β-roll fold might be masking a weaker α-helical signature, we examined the coevolved amino acid pairs identified by our approach. Twenty-one amino acid positions in the CTD formed only coevolved pairs corresponding to the β-roll fold, while positions exclusively forming coevolved pairs corresponding to the α-helical fold numbered only four (Fig. 5a).

To weaken the coevolutionary signal corresponding to the β-roll fold, we changed all 21 positions in the MSA to alanine, the mutation of choice for perturbing structure[43], except for the sequence of the Variant (Fig. 5b); positions forming different contacts in the two folds were left unchanged. From this modified MSA, AlphaFold2 predicted a ground state α-helical structure consistent with our coevolutionary predictions (Fig. 5b, Supplementary Fig. 14). The secondary structures

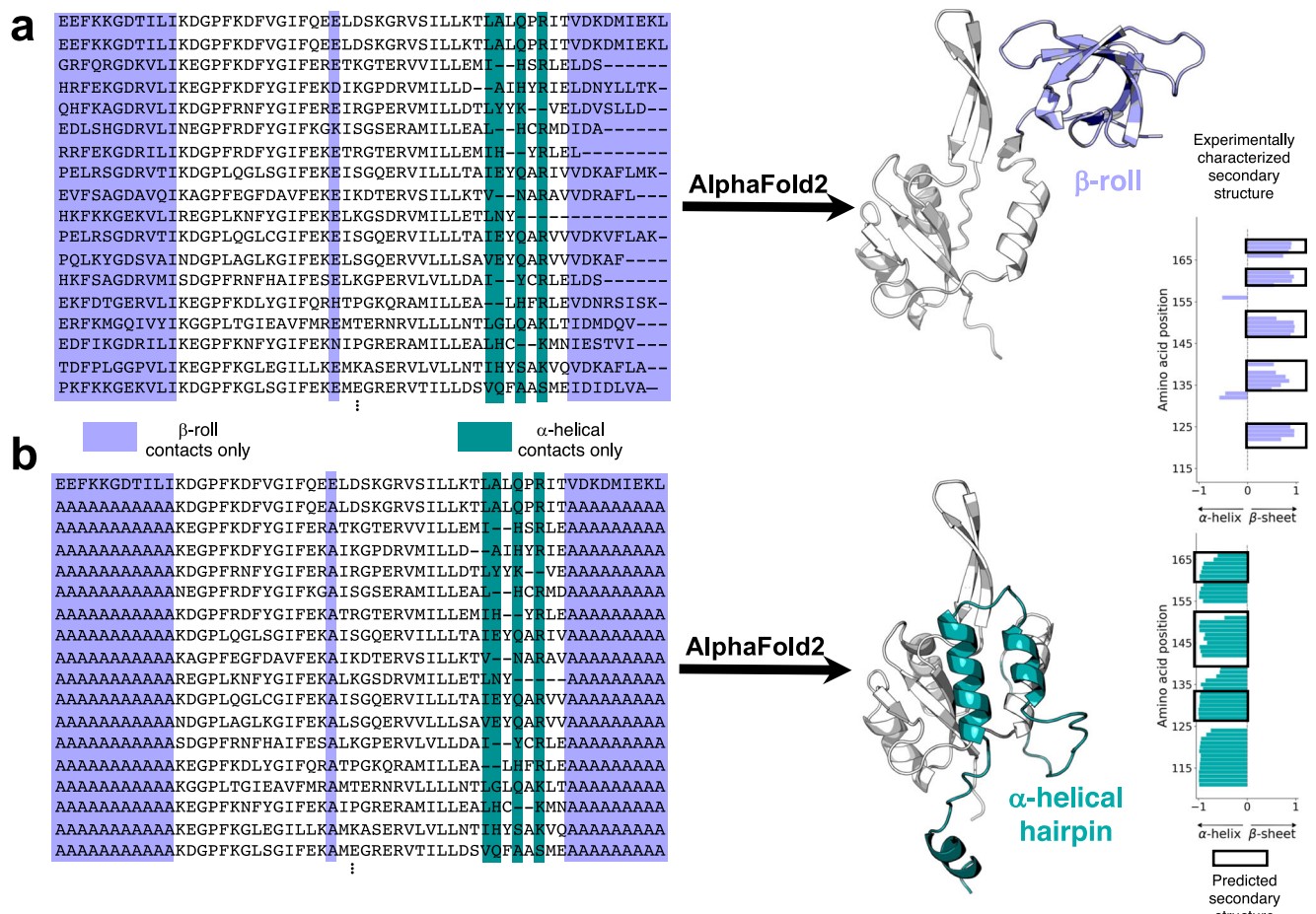

**Fig. 5 | AlphaFold2 successfully predicts two conformations of a candidate sequence without experimentally determined structures. a** A NusG N-terminal (NGN) fold (light gray) and a C-terminal β-roll fold (lavender) are predicted from a deep input MSA (region corresponding to the CTD shown). Predicted β-sheets in the C-terminal domain that agree closely with the β-sheets predicted from nuclear magnetic resonance experiments are shown with black boxes surrounding lavender bars. **b** A NusG N-terminal (NGN) fold (light gray) and a C-terminal α-helical hairpin fold (teal) are predicted from a modified input MSA in which columns predicted to form only β-roll contacts are changed to alanine. Predicted α-helices in the C-terminal domain that agree with the α-helices predicted from nuclear magnetic resonance experiments are shown with black boxes surrounding teal bars. Protein structures were generated with PyMOL[80]. Source data are provided as a Source Data file.

of both CTDs have high prediction confidences (pLDDT scores), except for the most C-terminal helix in the α-hairpin conformation (Supplementary Fig. 15a, b). RoseTTAFold2[44] predicted a similar helical conformation within 0.6 Å RMSD of the AlphaFold2 prediction when our modified MSA was inputted (Methods), confirming that alternative protein folds can be predicted by masking coevolutionary information in MSAs[43].

Both predicted conformations are consistent with amino-acid-specific secondary structure predictions calculated from nuclear magnetic resonance assignments[20] (Fig. 5a, b). Furthermore, without suppressing the strong β-roll coevolutionary signature, AlphaFold2 consistently predicted the β-sheet fold regardless of input MSAs and use or absence of templates. The α-helical CTD conformation was also missed by RoseTTAfold[1] and RGN2[45], an MSA-independent deep learning method that outperforms AlphaFold2 on orphan protein sequences (Supplementary Fig. 13). Importantly, masking coevolutionary signals in the experimentally characterized single-folding NusG protein from *Escherichia coli* resulted in an AlphaFold2 prediction of an unfolded CTD rather than an α-helical one (Supplementary Fig. 16a, b). Together, these results demonstrate that the coevolved contacts identified by our approach guided AlphaFold2 to predict the correct alternative conformation of an experimentally confirmed fold switcher.

## Not all AlphaFold2-generated fold-switch predictions have obvious coevolutionary signatures

We wanted to see if other MSA modifications have caused AlphaFold2 to produce fold-switched structures with strong coevolutionary support, like the NusG variant predicted here (Fig. 5a, b, Supplementary Fig. 14). Recently, four different fold-switching events have been predicted blindly using ColabFold[46], an efficient implementation of AlphaFold2 that generates comparable structure predictions: three in *E. coli* Adenylate Kinase (AK) and one in DsbE, an oxidoreductase from *Mycobacterium tuberculosis*. The first three were generated by masking coevolutionary signals within AK's MSA[43], the fourth by inputting a small cluster of sequences similar to DsbE[47]. Interestingly, our approach did not identify strong coevolutionary signatures for any of these four predictions (Fig. 6a, b, Supplementary Fig. 17), especially DsbE, whose putative fold-switched state has the largest number of higher order contacts. While these recent AlphaFold2 predictions may be fold switchers, they remain to be confirmed experimentally.

## Blind predictions of known fold switchers

Taking a more conservative approach to blind predictions of fold switchers, we tested an alternative strategy that avoids input MSA modification and cross-validates predictions by dual-fold coevolutionary signatures (Fig. 7a). Hypothesizing that different coevolutionary

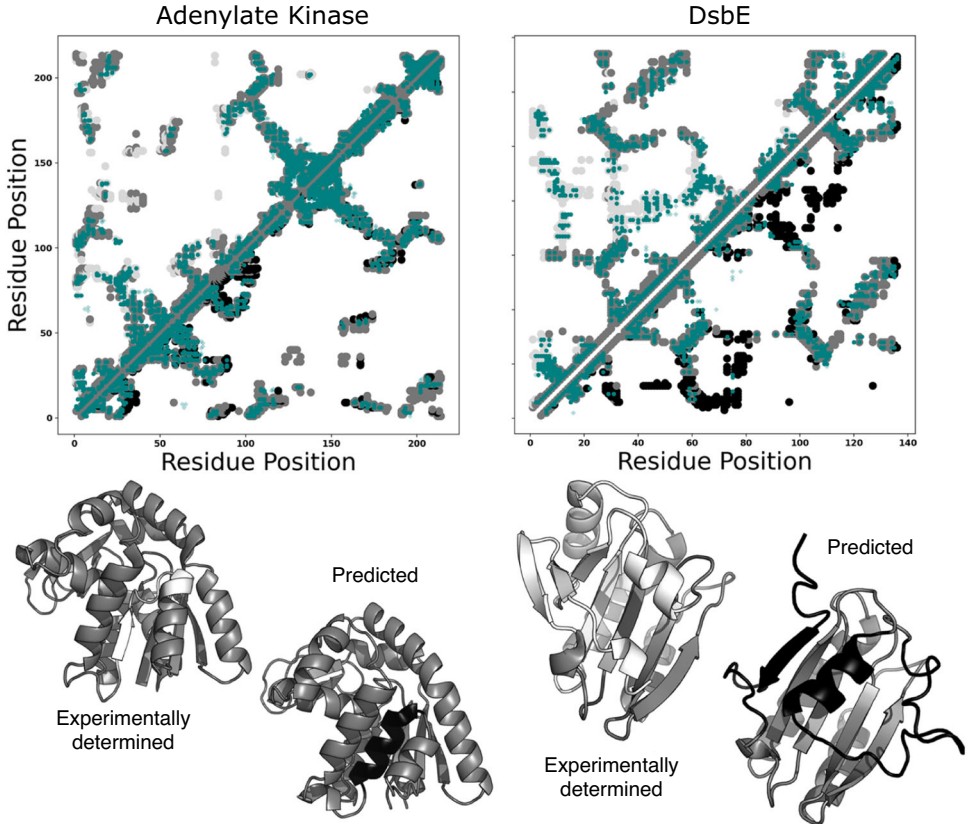

**Fig. 6 | Some AlphaFold2 fold-switch predictions based on modified multiple sequence alignments (MSAs) lack strong coevolutionary signatures.** Contact maps of Adenylate Kinase (left) and DsbE (right) show the experimentally determined structure on the top diagonal and the AF2-predicted fold switched structure on the bottom. Many predicted coevolved contacts (teal) overlap with contacts unique to the experimentally determined structures (light gray), but few overlap with contacts unique to the alternative structures predicted by AlphaFold2 (black). Structures of both sets of conformations are shown below their respective contact maps. Medium gray regions are common to both folds; white/black correspond to experimentally determined/AF2 prediction. PDB IDs for experimentally determined structures are 4AKE, chain A and 1LU4, chain A, for adenylate kinase and DsbE, respectively. Figure 1 provides an explanation of the dual-fold contact maps used here. Source data are provided as a Source Data file.

inference methods may favor different conformational states of fold-switching proteins, we compared three-dimensional structures generated by ColabFold[46]–a more efficient implementation of AlphaFold2–and ESMFold[3], a highly efficient computational method recently used to predict the structures of >600,000,000 proteins. While ColabFold infers residue-residue contact patterns from MSAs, ESMFold predicts contacts from single sequences using a large language model. Structural differences between the models produced by these two methods do not necessarily indicate fold switching, especially since ESMFold predictions can be less accurate than AlphaFold2, and by extension, ColabFold[3]. Thus, we cross-validated these two predicted structures with coevolved contacts inferred from ACE. We reasoned that if ACE-predicted contacts overlapped with the uniquely folding regions of both structures, they were likely both correct. Importantly, this approach is more efficient than previously proposed methods that modify the MSA inputs to AlphaFold2, which require several[47]–and sometimes many[43]–ColabFold runs on multiple modified MSAs. By contrast, our approach involves one ColabFold run on an unmodified MSA and one ESMFold run on a single sequence. Furthermore, this approach leverages the information gained from our dual-fold MSAs without using them for direct structural inference, which would likely be impeded by their suboptimal levels of experimentally uncharacterized contacts, many of which are likely to be noise (Supplementary Fig. 11b).

This blind predictive approach successfully identified fold switching in 13/56 known fold switchers (23%) with zero false positives. Successes are subdivided as follows. Category 1 comprises seven proteins with two correctly predicted conformations both corroborated by our coevolutionary pipeline (Fig. 7b, Supplementary Fig. 18). Figure 7b highlights MinE[48]–a bacterial protein whose fold switching fosters cell division–and *Entamoeba histolytica* calcium-binding protein-1, whose domain-swapped conformation may limit its target binding specificity[49]. Importantly, many of MinE's unobserved contacts correspond to its experimentally observed homodimeric interface. Category 2 comprises six other proteins for which only one conformation was predicted, but persisting coevolutionary signatures suggest a correct alternative conformation (Fig. 7c, Supplementary Fig. 19). For instance, bacterial PapA has a domain-swapped β-sheet (black) that fosters formation of large protein assemblies known as pili, which play a critical role in mediating bacterial adhesion to human urinary tracts[50]. Furthermore, the initiator protein RepE forms a monomeric and dimeric state with distinct conformations and functions: the monomeric form functions as a replication initiator, the dimer as a repressor[51]. Importantly, applying this approach to 181 expected single folders yielded no predicted fold switchers. In 22 cases ColabFold and ESMFold predicted different conformations in at least one protein region, but none of them were corroborated by coevolved contacts inferred by ACE. Thus, this predictive approach appears to be a reliable way to blindly predict fold-switching proteins. Although it will miss many true fold switchers, its low false positive rate (0% in this instance) suggests that the putative fold switchers it identifies will likely be correct.

## Discussion
Although globular proteins are generally observed to assume single unique folds, an increasing number can switch between distinct sets of

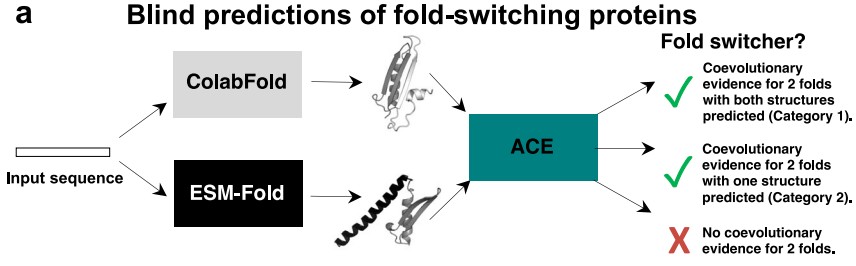

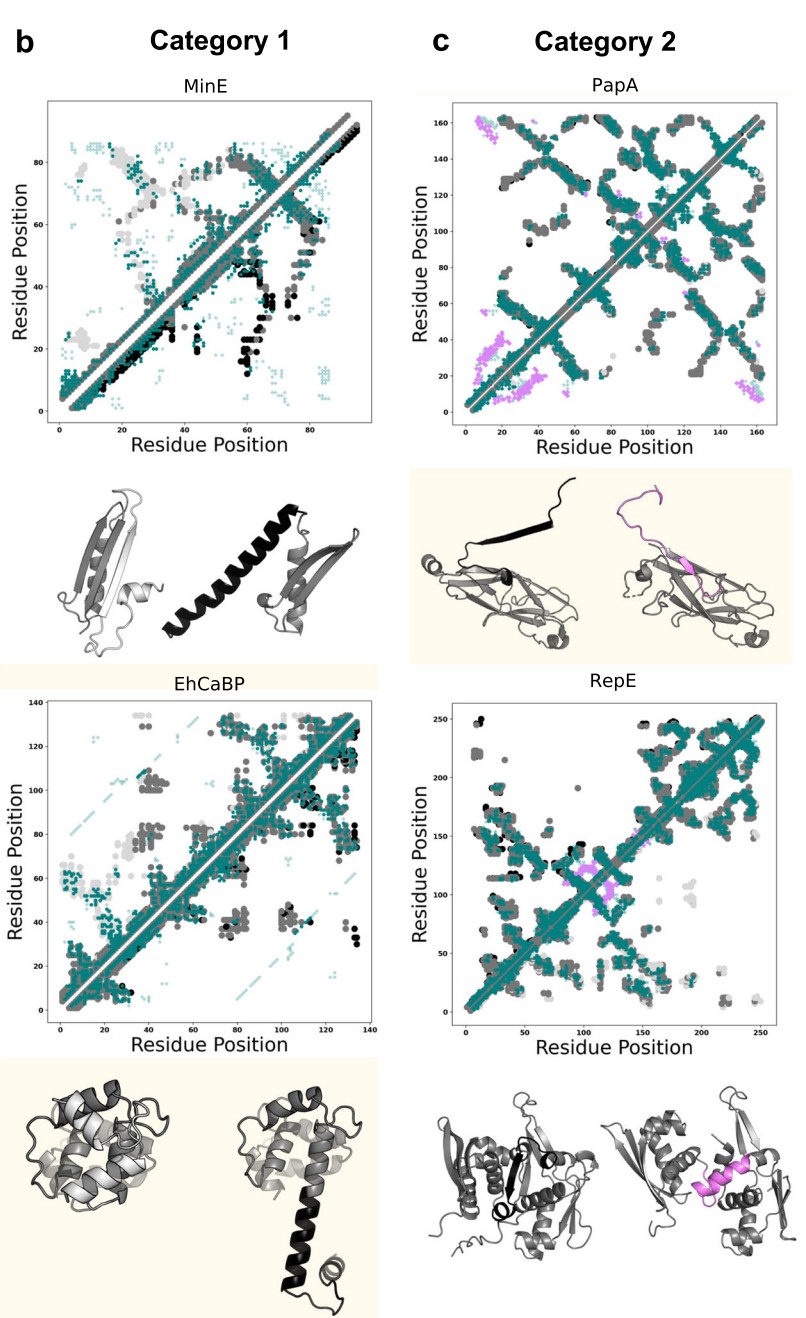

stable secondary and tertiary structure. These fold-switching proteins facilitate cancer progression[52], foster SARS-CoV-2 pathogenesis[53], fight microbial infection[34], and more[16]. The biological importance of many fold-switching proteins suggests that they may have been selected to assume two folds[54]. By running well-developed coevolutionary analysis methods[31,36,37] on many sets of unrelated protein superfamilies and

subfamilies, we identified statistically significant coevolutionary signals corresponding to two folds of 56 diverse fold-switching proteins. Although coevolutionary signals for alternative protein folds have been proposed previously for a small number of proteins[17,35,43], this work systematically identifies their origins (shallow subfamily-specific MSAs) and provides a biological rationale: dual-fold coevolutionary

**Fig. 7 | Blind predictions of fold-switching proteins. a** Blind predictions are performed by using ColabFold and ESMFold to each predict a structure of an amino acid sequence. ACE predicts coevolved residue pairs using the two predicted structures as references. The predicted structures are compared. Different structure predictions both consistent with coevolutionary predictions fall into Category 1 (**b**). Examples include the cell division protein MinE and the EF-hand protein EhCaBP. Similar structure predictions with coevolutionary evidence fall into Category 2 (**c**). Examples include the bacterial pilin protein PapA and the DNA replicase, RepE. For Figures (**b**) and (**c**), contact maps are shown above structures predicted by ColabFold (fold-switching regions light gray) and ESMFold (fold-switching regions black). Predicted contacts are teal. In (**c**) ColabFold and ESMFold predict the same conformation. Predicted contacts corresponding to the experimentally characterized alternative conformation are light purple. Structurally conserved protein regions/common contacts are medium gray. Although all proteins are presented as monomers for simplicity, MinE forms a dimer and PapA forms large oligomers. Figure 1 provides an explanation of the dual-fold contact maps used here. Source data are provided as a Source Data file.

signals arise from the sequences of protein subfamilies populated by fold-switching proteins rather than the superfamilies often dominated by single-proteins. These signals were then leveraged to (1) correctly predict two experimentally consistent conformations of a candidate protein with <30% sequence identity to its homologs with solved structures and (2) blindly predict fold switching of 13/56 proteins with zero false positives.

The widespread selection of proteins with two distinct structures indicates that fold switching (1) confers evolutionary advantage and (2) is a fundamental biological mechanism. These results, coupled with the difficulties associated with experimentally characterizing fold switchers[8,55], suggest that fold-switching proteins may be more abundant than currently realized. Accordingly, recent experimentally confirmed predictions suggest that over 3500 proteins in the NusG transcription factor family of ~15,500 proteins switch folds[20]. Furthermore, since subfamily MSAs have also been used to infer other protein properties[56,57], ACE might successfully extend beyond fold switchers to other forms of structural heterogeneity, such as allostery, which previous coevolutionary approaches have predicted with some success[58,59].

The observed prevalence and biological relevance of fold-switching proteins underscore the need to develop computational methods that reliably predict more. Although state-of-the-art predictive algorithms have revolutionized protein structure prediction[1,2,60], they systematically fail to predict protein fold switching[17,20]. Here, we suggest a computationally efficient pipeline to predict fold switching blindly. Although its 23% true positive rate is modest, its extremely low false positive rate (0/181), suggests that its fold switch predictions will likely be reliable. ACE was the key step in eliminating false positives. ColabFold and ESMFold predicted structural differences in 22/181 single folders, but none of these structural differences were supported by dual-fold coevolution. Thus, ACE not only demonstrates that evolution has selected for many fold-switching proteins but also can be used to cross-validate blind predictions of fold-switching proteins.

We expect that applying our blind predictive approach to thousands of sequences will yield numerous predicted fold switchers, many of which will be bona fide. The next challenge will be experimentally testing predictions. Most fold-switching events are triggered by external stimuli[15], and the triggers are often not obvious. For instance, RfaH was identified as a fold switcher in 2012[61]. Seven years later, the triggers of its reversible α-helix to β-sheet transition were reported[62]: binding both RNA polymerase and a specific DNA sequence, called *ops*. Screening for such non-obvious triggers will likely be difficult, but other fold switchers with simpler triggers, such as small molecules[63] or pH ref. [8], could potentially be identified through comprehensive screens. Furthermore, high-throughput structural screens for folds-witching need to be developed. Currently, no generalizable screens are available, though methods such as hydrogen-deuterium exchange mass spectrometry can identify slow conformational changes[64] and may therefore be used to screen for fold-switching, which occurs on the order of tens of milliseconds[65], seconds[66], or longer[67]. Circular dichroism can also screen fold switchers that undergo large shifts from α-helix to β-strand or vice versa[20].

Our findings lay the groundwork for a more functionally complete picture of the proteome by capturing dual-fold coevolutionary signatures of fold-switching proteins from their genomic sequences. In addition to developing a computational pipeline that blindly predicts fold switchers, we show that AlphaFold2 can be biased to predict two folds from one amino acid sequence. The key to this approach was suppressing the strong coevolutionary signature of the dominant β-roll fold, allowing AlphaFold2 to detect weaker α-helical signals from the amino acid sequence of a fold-switching NusG protein with low sequence identity to its PDB homologs. Importantly, the algorithm predicted no such signals from a single-folding NusG. This result confirms that dual-fold coevolutionary signals are present in a fold-switching NusG protein, but not in its single-folding homolog. On a cautionary note, running AlphaFold2 on the shallowest *E. coli* RfaH MSA used in our coevolutionary analysis yielded a nonsensical prediction with high confidence (ranked 0): a CTD with mixed α-helix and β-sheet character (Supplementary Fig. 20). Thus, we interpret high-confidence AlphaFold2 models inferred from modified MSAs with caution and run our blind prediction pipeline on full MSAs rather than modified ones.

Additional technical advances are needed to predict protein fold switching more reliably. First, coevolutionary signatures of fold switching must be distinguished from noise or true contacts arising from other phenomena, such as multimerization (e.g., MinE dimeric interface, Fig. 7b). Second, dual-fold contacts must be correctly separated into their two respective folds without prior knowledge of both conformations, on which we rely here. Nevertheless, ColabFold and ESMFold predictions captured the two distinct states of six fold-switching proteins and partially predicted both folds of a seventh. All seven sets of predictions were consistent with both sets of contacts inferred from ACE, giving us confidence that the blind predictive approach we developed will successfully predict some fold-switching proteins from whole genomes. Third, dual-fold coevolution must be predicted reliably. Our approach works only on sequences for which sufficiently deep MSAs can be generated. As a result, fold switching could not be predicted in 35% of the sequences in our initial dataset. Nevertheless, the rapid growth of diverse sequenced proteins[68], recent advances in deep learning[69,70], and increasingly accessible computational resources leave us optimistic that these challenges will be overcome.

## Methods

### MSA generation

Fold-switching protein sequences were used as inputs for jackhmmer[71,72] to generate MSAs after searching the Uniref90[68] release from January 2021. To achieve optimal MSA depths, multiple searches with -incE and -incdomE thresholds set to the same value ranging from $10^{-1}$ to $10^{-250}$ were performed in increments of $10^{-3}$. We then searched for the deepest MSA in this range with a maximum of 60,000 sequences. Each jackhmmer run was iterated until the MSA converged or until 10 iterations had occurred.

### MSA preparation

To generate subfamily MSAs, distantly related sequences were pruned from deep superfamily MSAs using hhfilter[73]. This software filters alignments by QID, pairwise sequence identity between the query sequence used to generate the MSA and each subsequent sequence

within it. Subfamily MSAs of varying depths were generated with QID thresholds ranging from 1% to 50% in increments of 1%. All MSAs−both superfamily and subfamily−were prepared for coevolutionary analysis by removing any sequences with >25% gaps and then filtering any columns with >75% gaps.

## Coevolutionary analysis

Prepared MSAs from each protein family were used as separate inputs into both GREMLIN[36,37] and MSA transformer[31], each run with default parameters. Typically, the number of coevolved amino acid pairs retained from each run from both programs is $3L/2$[37,74], where $L$ is length of the target protein. Here, a superposition of all coevolutionary predictions is created and the most probable $15L/2$ amino acid pair predictions are retained. The superposition reports the average z-score of each amino acid pair across all subfamilies. Contacts are categorized as follows. Dominant fold: unique contacts corresponding to the experimentally determined structure that overlaps most with predicted contacts from the deepest MSA; Alternative fold: unique contacts corresponding to the other experimentally determined structure; Common: predicted contacts overlapping with experimentally determined contacts shared by both folds; Unobserved: predicted contacts that do not overlap with any experimentally determined contacts. In all cases, overlap was defined as being with +/−2 residues of crystallographic (or predicted) contacts.

## Noise filtering

All predicted contacts generated from the original MSA and the subfamily MSAs were superimposed onto a single contact map. These predictions were clustered using a density-based algorithm (DBSCAN)[75] that efficiently identifies structure in datasets with arbitrarily shaped clusters. The main criteria for defining whether a point belongs to a cluster is how many other points are close. The eps parameter defines a radial distance from a core point and points within that radius are clustered. All points included in the cluster were then used as new core points to search for additional points within the eps. Clusters were iteratively built in this way until the entire dataset is clustered. The minimum number of points to define a cluster in this work is 3. The sparsest points in the dataset were then defined as noise and eliminated from the dataset to produce the final, densest set of filtered predictions. The eps value is optimized for each set of contacts calculated from experimentally determined or predicted protein structures using a receiver operating characteristic curve, where the optimal value's first derivative >1, corresponding to more true positives gained by increasing the eps value, but the successive value's first derivative <1, corresponding to more false positives gained by further increasing the eps value. True positives are defined as being within +/−2 residues of crystallographic (or predicted) contacts. However, eps values could not be so stringent that fewer contacts were returned than from the original run on deep MSAs.

## Statistical tests

p-values were calculated using the one-tailed hypergeometric test (also known as Fisher's exact test) to evaluate the significance of the additional structural information obtained from the subfamily alignments, as described by:

$$\sum_{i=0}^{N_{\mathrm{noise}subfamilies}} \frac{\binom{N_{\exp}-N_{\mathrm{pred}superfamily}}{N_{\mathrm{pred}subfamilies}+i}\binom{N_{noise}-N_{\mathrm{noise}superfamily}}{N_{\mathrm{noise}subfamilies}-i}}{\binom{(N_{\exp total}-N_{\mathrm{pred}superfamily})+(N_{noise total}-N_{\mathrm{noise}superfamily})}{N_{\mathrm{pred}subfamilies}+N_{\mathrm{noise}subfamilies}}} \quad (1)$$

where $N_{\exp}$ is the total number of unique experimentally determined contacts from both conformations of a fold-switching protein, $N_{\mathrm{pred}superfamily}$ is the number of unique contacts correctly predicted by GREMLIN and MSA transformer on the superfamily MSA only, $N_{\mathrm{pred}subfamilies}$ is the number of unique contacts predicted by GREMLIN

and MSA transformer on all subfamily MSAs excluding those also predicted from the superfamily, $N_{noise}$ is $L^2 - N_{\exp}$, where $L$ is the maximum sequence length of an experimentally determined structure, $N_{\mathrm{noise}superfamily}$ is the number of unique contacts incorrectly predicted by GREMLIN or MSA transformer on the superfamily MSA only, and $N_{\mathrm{noise}subfamilies}$ is the number of unique contacts incorrectly predicted by GREMLIN or MSA transformer on all subfamily MSAs, excluding those also predicted from the superfamily. Epps-Singleton tests on distributions in Fig. 3c were performed using the scipy stats module on non-dominant contacts from single-fold (181) and fold-switching (58) proteins.

## Single fold dataset

A comparison dataset of monomeric proteins was constructed to compare to the 56 fold-switching proteins. These proteins were taken from the CAMEO dataset from[37], excluding complexes and de novo proteins. These 181 monomers were then run through the pipeline and non-dominant contacts (all contacts not corresponding to experimentally determined contacts) associated with all 181 monomers were compared to the non-dominant (noise+alternative) contacts associated with the 56 fold switchers.

## Structure predictions

Structure predictions of Variant 5 were performed by AlphaFold2.1.2 both with templates deposited in the PDB by 4/20/22 and without templates and both with MSAs generated from the standard pipeline (Uniref90[68], MGnify[76], and MMseqs2[77] (BFD clust)) and the shallowest MSA generated from our approach. In all four runs, only the β-roll fold was predicted in the five top-scoring models (Supplementary Fig. 12). The α-helical fold was predicted by modifying MSA columns that our pipeline predicted to form only β-roll contacts. These columns corresponded to amino acids in the 100−168 range, and the deepest MSA generated by our approach was modified by mutating these columns to alanine. AlphaFold2.1.2 was run on this modified MSA without templates. The α-helical conformation did not result from alanine substitution: mutating the same MSA columns to their corresponding amino acid in Variant 5's sequence instead of alanine yielded the same prediction.

When RoseTTAFold2[44] was run on the sequence of the NusG Variant using default settings, it predicted structures whose CTDs assumed the β-roll fold only. Upon inputting the alanine-substituted Uniref90 MSA used to bias AlphaFold2 to predict the Variant's helical CTD conformation, RoseTTAFold2 also predicted the structures with same helical conformation. The overall RMSD (NTD + CTD) of these structures was within 0.6 Å of the helical AlphaFold2 prediction (Fig. 5). In addition to the modified input MSA, 15% of the Variant's sequence was masked at random, and 256 sequences were randomly selected as input MSAs for 16 independently predicted models, 6 of which had helical CTDs and the remaining 10 had β-roll CTDs. As a control, we ran RoseTTAFold2 using the same parameters while inputting the Variant's Uniref90 MSA without alanine substitutions. Inputting this unmodified MSA yielded 16/16 predictions with β-roll CTDs. Thus, the alanine substitutions in our input MSA successfully biased both RoseTTAFold2 and AlphaFold2 to predict experimentally consistent α-helical CTDs of the NusG Variant. All RoseTTAFold2 runs were performed using Sergey Ovchinnikov's publicly available Colab notebook: https://colab.research.google.com/github/sokrypton/ColabFold/blob/main/RoseTTAFold2.ipynb

The standard RoseTTAfold pipeline (https://robetta.bakerlab.org) was used to predict three-dimensional structures of Variant 5 with the shallowest MSA generated from our pipeline.

The RGN2 Colab notebook

(https://colab.research.google.com/github/aqlaboratory/rgn2/blob/master/rgn2_prediction.ipynb) was run on the sequence of Variant 5 with standard parameters. The sequence of Variant 5 is:

```
MESFLNWYLIYTKVKKEDYLEQLLTEAGLEVLNPKIKKTKTVRNKKKEVI
DPLFPCYLFVKADLNVHLRIISYTQGIRRLVGGSNPTIVPIEIIDTIKSRMVD
GFIDTKSEEFKKGDTILIKDGPFKDFVGIFQEELDSKGRVSILLKTLALQPRI
TVDKDMIEKLHN.
```
**.** Experimentally determined secondary structures were taken from[20]. ESMFold predictions were generated with a local install of the ESMFold software (https://github.com/facebook research/esm).

## Blind prediction pipeline
Structures of each experimentally confirmed fold switcher were predicted by independently inputting their sequences into ColabFold and ESMFold run with standard parameters. Resulting structures were inputted into ACE to improve noise filtering. Predicted structures that both differed in the fold-switching regions and were each corroborated by dual-fold coevolution were classified as Category 1. In 7/8 cases, both predicted structures matched experimentally determined protein conformations; in the 8th (RfaH), one matched the α-helical conformation, while the other was a mixture of helix and β-sheet in the fold-switching region. Cases in which ColabFold and ESMFold predicted the same conformation, but substantial signal corresponding to the alternative fold remained present were classified as Category 2. All cases with coevolutionary evidence for one conformation (i.e., no alternative contacts) were considered single folders. As a control, this procedure was also performed on 181 single-fold proteins; none of them showed coevolutionary evidence for an alternative protein conformation.

To determine whether ColabFold and ESMFold predictions of the same sequence had regions with different structures, secondary structure annotations of each PDB, by DSSP[78], were compared one-by-one, position-by-position. This approach allowed us to quantitatively assess the similarity of aligned secondary structures. A potential fold switcher was required to have a continuous region of at least 15 residues in which at least 50% of the residues showed α-helix ↔ β-sheet differences[79].

## Reporting summary
Further information on research design is available in the Nature Portfolio Reporting Summary linked to this article.

## Data availability
The coevolutionary plots for all 56 fold-switching proteins and predicted structures of the candidate NusG have been deposited on Github under accession code: https://github.com/ncbi/dual_fold_coevolution. PDB accession codes used in Fig. 1: 4KSO, chain A and 1T4Y, chain A. PDB accession codes used in Fig. 6: 4AKE, chain A and 1LU4, chain A. Chemical shifts from which secondary structure assignments were made are deposited in the BMRB with accession codes 51529 [https://doi.org/10.13018/BMR51429] (α-helical conformation) and 51428 (β-sheet conformation)[20]. Source data are provided as a source data file. Source data are provided with this paper.

## Code availability
Code used to generate the results reported in this manuscript can be found at: https://github.com/ncbi/dual_fold_coevolution.

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

## Acknowledgements

We thank Carolyn Ott, Loren Looger, Devlina Chakravarty, Yuri Wolf, Robert Best, Andy LiWang, Eugene Koonin, George Rose, Danielle and Jean Thierry-Mieg, and Nash Rochman for helpful discussions. This work utilized resources from the NIH HPS Biowulf cluster (http://hpc.nih.gov), and it was supported by the Intramural Research Program of the National Library of Medicine, National Institutes of Health (LM202011, L.L.P.).

## Author contributions

Conceptualization: L.L.P., J.W.S. Methodology: J.W.S., L.L.P. Software: J.W.S. Investigation: J.W.S., L.L.P. Data Curation: J.W.S., L.L.P. Visualization: J.W.S, L.L.P. Writing – original draft: J.W.S, L.L.P. Writing – review & editing: L.L.P. Supervision: L.L.P. Project administration: L.L.P. Funding acquisition: L.L.P.

## Funding

## Competing interests

The authors declare no competing interests.
