## [Peer Review File · Nature Communications]

Evolutionary selection of proteins with two foldsEditorial Note: This manuscript has been previously reviewed at another journal that is not operating a transparent peer review scheme. This document only contains reviewer comments and rebuttal letters for versions considered at Nature Communications. Mentions of prior referee reports have been redacted.

REVIEWER COMMENTS

Reviewer #3 (Remarks to the Author):

As I noted in my original review - this is an exciting and timely manuscript. The revised manuscript has incorporated my suggestions and addressed all of my concerns. The current version has added a section about potential use and generalization of this approach. I really like this - but encourage the authors to make sure not to let it bury the overall evolutionary implications of their study. Please note again that I am not a computational biologist and can not comment critically on that aspect of the manuscript.

I have one minor correction:

Line 293-294 should read

'Importantly, masking coevolutionary signals in the experimentally characterized single-folding NusG protein from Escherichia coli resulted in an AlphaFold prediction of an unfolded CTD rather than an α -helical one (Supplementary Figure 7).'

and not

'Importantly, masking coevolutionary signals in the experimentally characterized single-folding NusG protein from Escherichia coli resulted in an unfolded CTD rather than an α -helical one (Supplementary Figure 7).'

Reviewer #4 (Remarks to the Author):

The revision has improved the story by incorporating more analysis and example discussion. Nevertheless, some concerns remain.

1, it is interesting that the authors claim that "Superfamily as now been defined in the Introduction", but I found the appearance of the word "Superfamily" in the introduction part has remained identical to the previous version.

2, I am not fully convinced that step-by-step pruning strategy works better than clustering. And clustering could be done very fast.

3, the answer to my concern on "The same amino acids in a protein may involve different residue-residue contacts of different conformations " has missed my point. I meant the amino acids involved in different contacts in two folds, not common.

4, it remains not clear from the methods how " Alternative fold contacts" are defined in the Methods part.

5, why RoseTTAFold crashed?

Referees' comments:

Reviewer #3 (Remarks to the Author):

As I noted in my original review - this is an exciting and timely manuscript. The revised manuscript has incorporated my suggestions and addressed all of my concerns. The current version has added a section about potential use and generalization of this approach. I really like this - but encourage the authors to make sure not to let it bury the overall evolutionary implications of their study. Please note again that I am not a computational biologist and can not comment critically on that aspect of the manuscript.

We thank the Reviewer for their constructive comments.

I have one minor correction:

Line 293-294 should read

'Importantly, masking coevolutionary signals in the experimentally characterized single-folding NusG protein from Escherichia coli resulted in an AlphaFold prediction of an unfolded CTD rather than an α -helical one (Supplementary Figure 7).'

and not
'Importantly, masking coevolutionary signals in the experimentally characterized single-folding NusG protein from Escherichia coli resulted in an unfolded CTD rather than an α -helical one (Supplementary Figure 7).'

We thank the Reviewer for this correction. We have updated the Manuscript accordingly.

Reviewer #4 (Remarks to the Author):

The revision has improved the story by incorporating more analysis and example discussion.

We thank the Reviewer for their positive comment.

Nevertheless, some concerns remain.

1, it is interesting that the authors claim that "Superfamily as now been defined in

the Introduction", but I found the appearance of the word "Superfamily" in the introduction part has remained identical to the previous version.

The Reviewer seems to have missed this definition in our previous revision. Introduction, paragraph 4, sentence 2: "MSAs from protein superfamilies (deep MSAs containing a large clade of diverse-yet-homologous sequences)". This change was tracked in the previous revision. We made it bold in this version to help the Reviewer identify it quickly while indicating that it has not changed since our last revision.

2, I am not fully convinced that step-by-step pruning strategy works better than clustering. And clustering could be done very fast.

The Reviewer originally suggested that we benchmark against a *bioRxiv* manuscript, Wayment-Steele et. al. 2022. (reference 48 in our manuscript) that uses sequence clustering to successfully predict two conformations of three experimentally characterized fold-switching proteins. We reference this manuscript in two sections of the Results: *Not all AlphaFold2-generated fold-switch predictions have obvious coevolutionary signatures* and *Blind predictions of known fold switchers*.

Benchmarking against this method, or trying other sequence clustering methods, is outside of the scope of this manuscript for four reasons.

First, the overall message of our paper will not change at all, regardless of the results. If clustering enhances our predictions, the subfamily MSAs with sequences similar the query sequence still provide coevolutionary information about the alternative conformation. Same if clustering doesn't enhance our predictions, as our current manuscript shows.

Second, as Reviewer 3 said, the additional information included in our previous revision is starting to diminish the overall message of the paper: fold-switching proteins have been selected by evolution to assume two different conformations, and this information can be leveraged to predict two folds from one sequence some of the time. Clustering sequences will not change this message, regardless of the outcome, but it could further bury the main points of the manuscript.

Third, using AF-cluster for our method would be quite computationally intensive. As noted in our previous response to the Reviewer, the clusters generated by AF-cluster generally do not contain enough sequences for our pipeline. To illustrate this, we

looked at the clusters it generated for RfaH, KaiB, and Mad2—the three proteins with two conformations successfully predicted using AF-cluster. For both RfaH and Mad2, none of the clusters had enough sequences to run our analysis (at least 5*length of the input protein sequence). Note that there were 224 (RfaH) and 106 (Mad2) clusters in total. For KaiB the sequence depth of only 1 cluster was sufficient for our analysis (out of 228). Again, as noted in our previous response, determining the optimal combination of smaller sequence clusters would be quite computationally intensive: 224! for RfaH, 228! for KaiB, and 106! for Mad2. An exhaustive combinatorial approach is not feasible, and developing a way to optimally combine clusters would require us to reengineer ACE. Out of curiosity, we ran ACE on the one KaiB sequence cluster deep enough for analysis. The noise of the prediction was 9x higher and the number of contacts it predicted correctly decreased by 9% compared to our pruning approach.

Finally, the AF-cluster manuscript has not yet been peer-reviewed. It seems imprudent to benchmark against a yet-unproven method that could change substantially before it is accepted for publication in a peer-reviewed journal.

3, the answer to my concern on "The same amino acids in a protein may involve different residue-residue contacts of different conformations " has missed my point. I meant the amino acids involved in different contacts in two folds, not common.

We apologize that our previous response was unclear. Amino acids involved in different residue-residue contacts in the two folds were not mutated to Alanine. We updated the manuscript accordingly: “positions forming different contacts in the two folds were left unchanged.”

4, it remains not clear from the methods how " Alternative fold contacts" are defined in the Methods part.

We thought this was important enough that we had defined it in Results, paragraph 4:

“Contacts are categorized as follows. Dominant fold: unique contacts corresponding to the experimentally determined structure that overlaps most with predicted contacts from the deepest MSA (light gray contacts in Figure 2b-d); Alternative fold: unique contacts corresponding to the other experimentally determined structure (black contacts in Figure 2b-d); Common: predicted contacts overlapping with experimentally determined contacts shared by both folds (gray contacts symmetric on both sides of the diagonal in Figure 2b-d);

Unobserved: predicted contacts that do not overlap with any experimentally determined contacts (readily visible in Figure 2c).”

We have now included a very similar statement in the Methods section on Coevolutionary analysis.

5, why RoseTTAFold crashed?

We successfully inputted our alanine-substituted MSA into RoseTTAFold2. It predicted 6/16 CTDs with ground-state helical folds. The RMSDs of all 6 structures were within 0.6Å of the structure predicted by AlphaFold2. This result confirms that alternative protein folds can be predicted by masking coevolutionary information in MSAs.

This finding is now mentioned in the Results and further discussed in Methods:

“When RoseTTAFold2 was run on the sequence of the NusG Variant using default settings, it predicted structures whose CTDs assumed the β -roll fold only. Upon inputting the alanine-substituted Uniref90 MSA used to bias AlphaFold2 to predict the Variant’s helical CTD conformation, RoseTTAFold2 also predicted the structures with same helical conformation. The overall RMSD (NTD+CTD) of these structures was within 0.6Å of the helical AlphaFold2 prediction (Figure 5). In addition to the modified input MSA, 15% of the Variant’s sequence was masked at random, and 256 sequences were randomly selected as input MSAs for 16 independently predicted models, 6 of which had helical CTDs and the remaining 10 had β -roll CTDs. As a control, we ran RoseTTAFold2 using the same parameters while inputting the Variant’s Uniref90 MSA without alanine substitutions. Inputting this unmodified MSA yielded 16/16 predictions with β -roll CTDs. Thus, the alanine substitutions in our input MSA successfully biased both RoseTTAFold2 and AlphaFold2 to predict experimentally consistent α -helical CTDs of the NusG Variant. All RoseTTAFold2 runs were performed using Sergey Ovchinnikov’s publicly available Colab notebook: <https://colab.research.google.com/github/sokrypton/ColabFold/blob/main/RoseTTAFold2.ipynb>”

When we inputted alanine-substituted MSAs (we tried several of different depths) into RoseTTAFold using the Robetta server, the server consistently returned an error. We do not have access to the error file and therefore cannot comment on the cause.